# Distinguishing Ideal and Non-Ideal Chemical Systems Based on Kinetic Behavior

**DOI:** 10.3390/e27010077

**Published:** 2025-01-16

**Authors:** Gregory Yablonsky, Vladislav Fedotov

**Affiliations:** 1Department of Energy, Environmental and Chemical Engineering, McKelvey School of Engineering, Washington University in St Louis, St. Louis, MO 63130, USA; 2Department of Information Systems, Chuvash State University, Moskovsky pr. 15, 428015 Cheboksary, Russia; fvh@inbox.ru

**Keywords:** non-ideal chemical systems, Marcelin–de Donde kinetic, one- and two-step mechanism, intersection point, CPE-extremum point, fingerprint

## Abstract

This paper focuses on differentiating between ideal and non-ideal chemical systems based on their kinetic behavior within a closed isothermal chemical environment. Non-ideality is examined using the non-ideal Marcelin–de Donde model. The analysis primarily addresses ‘soft’ non-ideality, where the equilibrium composition for a reversible non-ideal chemical system is identical to the corresponding composition for the ideal chemical system. Our approach in distinguishing the ideal and non-ideal systems is based on the properties of the special event, i.e., event, the time of which is well-defined. For the single-step first-order reaction in the ideal system, this event is the half-time-decay point, or the intersection point. For the two consecutive reversible reactions in the ideal system, A ↔ B ↔ C, this event is the extremum obtained within the conservatively perturbed equilibrium (CPE) procedure. For the non-ideal correspondent models, the times of chosen events significantly depend on the initial concentrations. The obtained difference in the behavior of the times of these events (intersection point and CPE-extremum point) between the ideal and non-ideal systems is proposed as the kinetic fingerprint for distinguishing these systems.

## 1. Introduction

A big problem in physical chemistry and chemical engineering is distinguishing the ideal and non-ideal chemical systems. Typically, the chemical ideality is verified assuming a weak interaction between the molecules of the composition under the condition that the solution is diluted enough.

The fundamental properties of the ideal closed chemical system are the uniqueness and stability of the chemical equilibrium composition at fixed amounts of the chemical elements and a fixed temperature. The chemical kinetics of the ideal chemical system are described by Mass Action Law (MAL) dependences. For this system, the uniqueness and stability of the equilibrium were first qualitatively proven by Zeldovich in 1938 [1,2]; from 1960 onwards, many researchers studied these problems and presented rigorous proofs of the uniqueness and stability of the equilibrium composition, such as Shapiro and Shapley [3], Aris [4,5], Horn and Jackson [6], Vol’pert and Khudyaev [7,8], and Gorban and Yablonsky [9,10]. It is essential that the equilibrium of the reversible complex reaction is the detailed equilibrium—i.e., for every step considered separately, the rate of the forward reaction equals the rate of the corresponding reverse reaction. Reviews of these results are available in books [11].

As for the non-ideal chemical isothermal system, in which the MAL is violated, the generalized Marcelin–de Donder (MDD) model was proposed in terms of non-ideal chemical potentials [12,13,14]. The uniqueness and stability of the non-ideal system were proven under special assumptions on the non-ideality coefficients [11]. For the non-ideal system, the principle of the detailed equilibrium is fulfilled as well.

In this paper, we are going to develop fingerprints for comparing and distinguishing the ideal and non-ideal systems assuming some simple chemical mechanisms as examples. For this purpose, some recent findings in formal chemical kinetics will be used, the so-called “joint kinetics” approach, i.e., the properties of the intersections of kinetic dependencies and transition regimes from the special initial conditions.

In [15], the effect of the conservatively perturbed equilibrium (CPE) was found, described, and studied in the closed ideal chemical system obeying MAL kinetics. The essence of the effect is the specific kinetic behavior of a chemical system in response to the replacement of the initial concentrations of one or several (but not all) reactants by the corresponding equilibrium concentrations at the same total amount of each chemical element and a fixed temperature. A distinctive feature of the CPE effect is the unavoidable appearance of a concentration extremum during relaxation to the final equilibrium. This is observed for the substance whose initial concentration is taken as the equilibrium one, as shown in Figure 1.

This extremum may be an instantaneous partial equilibrium of some steps of the chemical mechanism. In [16], the CPE effect was experimentally confirmed for the reaction of ethanol and benzyl alcohol with acetic acid to form esters in a batch reactor. The CPE effect provides new information about the detailed mechanism of this complex reaction.

This article discusses the possibility of observing this effect in non-ideal isothermal systems with the generalized Marcelin–de Donder kinetic law. The CPE phenomenon is studied for different types of non-ideal chemical potentials, provided that the uniqueness and stability of the chemical equilibrium composition are preserved. The case of ‘soft’ non-ideality is analyzed, when, for any reversible chemical reaction, the equilibrium constant for ideal chemical potentials is equal to the “non-ideal” equilibrium constant. As a result, the equilibrium composition of the non-ideal and ideal system must be identical. This creates additional requirements for non-ideal parameters that affect the reaction dynamics. The main goal of this work is to analyze, under the conditions of the CPE phenomenon, the dynamic properties of a kinetic model of the non-ideal system and compare it with a corresponding model of the ideal one.

## 2. Kinetic Model

A chemical reaction is described by a set of elementary steps, as follows:∑*_j_*α*_ij_* A*_j_* ↔ ∑*_j_*β*_ij_* A*_j_*,(1)
where α*_ij_* and β*_ij_*—stoichiometric coefficients of the *j*-th component in the *i*-th step and A*_j_*—initial components and reaction products. The kinetic model of this reaction in a closed gradientless isothermal reactor is described by the system of ordinary differential equations [11,17,18,19,20], as follows:*A_j_*′ = ∑*_i_*(β*_ij_* − α*_ij_*)*r_i_*, *A_j_*(0) = *A_j_*_0_,(2)
where *A_j_* = *A_j_*(*t*)—reactant concentrations, mol/L; *t*—time, s; *r_i_*—the rate of any step reaction (ideal or non-ideal) that satisfies the natural requirement (*r_i_* = 0 at *A_j_
*= 0), mol/(L s); *r_i_ = r_i_^+^* − *r_i_^−^,* where *r_i_^+^* and *r_i_^−^* are the rates of the forward and reverse reactions, respectively, mol/(L s); and *A_j_*_0_—initial concentrations, mol/L. In model (2), at any time, the conservation law is satisfied, i.e., the system is conservative, as follows:∑*_j_ A_j_*(*t*) = ∑*_j_ A_j_*_0_ = 1.(3)

In our paper, the difference between the ideal and non-ideal systems is expressed in terms of chemical potentials.

In the ideal kinetics of the MAL, the rate of a reversible isothermal reaction [1,11,17,18,19,20] is equal to *r_i_* = *r_+__i_* − *r*_−*i*_, where *r_+__i_
*= *k_+__i_
*∏*A_j_
*^α*ij*^, *r_−__i_* = *k_−__i_*∏*A_j_
^βij^*— the rate of forward and reverse reactions, respectively, 1/s.

In the non-ideal kinetics of MDD, which is a generalization of the ideal kinetics of the MAL, the rate of any step of a reversible isothermal reaction [9,10,11,12,13,14,18,19,20] is the following:*r_i_* = *r_i_*^0^[exp(∑*_j_* α*_ij_*μ*_j_*) − exp(∑*_j_* β*_ij_*μ*_j_*)], *i* = 1, 2, …,(4)
where *r_i_*^0^ > 0—the rate, forward or reverse, under the equilibrium conditions, mol/(L s) and µ*_j_*—the pseudo-chemical potential (hereinafter referred to as the potential) of the reactant A*_j_*, dimensionless, as follows:μ*_j_*= μ^0^*_j_* + ln *A_j_
*+ *g_j_*, *j* = 1, 2, …,(5)
where μ^0^*_j_* > 0—constants; *g_j_*—correction terms for the non-ideality of reactants (*g_j_* = 0 for the ideal system); µ*_j_* = µ**_j_*/*RT*, µ**_j_*—the chemical potential of the reactant A*_j_*, J/mol; *R*—the universal gas constant, J/(mol °K); and *T*—temperature, °K.

The goal of this paper is to find the difference between the ideal and non-ideal cases based on kinetic behavior.

Let us assume that the non-ideal correction terms *g_j_* = ∑*a_jk_A_j_
*are linear regarding the concentrations of reactants, where *a_jk_* are non-ideality coefficients of reactant *A_j_*, *k* = 1, 2… In an ideal system, all coefficients of non-ideality are equal to 0, *a_jk_* = 0, and *g_j_* = 0. In a particular case, the coefficients of non-ideality and the corresponding correction terms are equal (‘uniform’ non-ideality). In a general case, *a_jk_* may have any values, and at least one of the correction terms may be not equal to 0 (‘non-uniform’ non-ideality). In our calculations, such values of *a_jk_* are chosen that the non-ideal constituent is at least about 20% of the ideal one.

The stability of the system (2)–(5) is determined by the eigenvalues λ*_j_* of the Jacobian matrix ***J*** = ***J***_γ_***J***_μ_ near the equilibrium, where ***J***_γ_ = (β*_ij_* − α*_ij_*)—the matrix of stoichiometric coefficients related to the scheme of transformations; ***J***_μ_ = (μ*_jk_*) is a matrix of potentials which describes the non-ideality of reactants. It was proved [5,6,7,8] that, if the matrix of partial derivatives (Jacobi matrix) for potentials ***M*** = (*∂*µ*_j_*/*∂**A_k_*) is symmetric and positively definite, i.e., non-negative, all its main minors are non-negative, det(***M***) ≥ 0, then the equilibrium is unique and stable globally. Any violation of these conditions can lead to incorrect conclusions about the multiplicity of equilibria or the instability of the single equilibrium in the closed system [21,22].

Chemical dynamics are characterized by relaxation times [23,24,25,26]. Generally, the relaxation time reflects the dynamics of the system at any moment in time (near and far from the equilibrium) and depends on the moment of the occurrence of the event (intersection, extremum, etc.) and the initial conditions of the reaction *t* = *t*(*A_j_*, *A_j_*_0_). As is well known, a set of eigenvalues λ*_j_* (linear relaxation times) present the robust characteristic of relaxation. Typically, the slowest relaxation is characterized by the linear relaxation time, τ = 1/min|Re λ*_j_*|, where Re is the real part of the eigenvalue λ*_j._*

As is indicated above, the relaxation characteristics of typical chemical mechanisms will be studied in detail, comparing the ideal and non-ideal cases and distinguishing them. Our analysis will be focused on different events of interest, i.e., the intersection of kinetic curves for the single reaction and the CPE phenomenon for the two-step reaction.

## 3. Linear One-Step Mechanism

In this case, the simplest mechanism is the following one-step isomerization reaction:A ↔ B,
where A and B are reactants, the initial reactant and product. The corresponding kinetic model is written as follows:*A*′ = −*r*_+_ + *r*_−_, *B*′ = *r*_+_ − *r*_−_, *A*(0) = *A*_0_,
where *A* = *A*(*t*) and *B* = *B*(*t*)—reactant concentrations, *t*—reaction time; and *r_+_*, *r_−_*—reaction rates, forward and reverse, respectively. In this system, the mass conservation law is fulfilled *A*(*t*) +* B*(*t*) = 1. It does not depend on whether the system is ideal or non-ideal.

The special kinetic event for this simple system is an intersection of the kinetic dependencies *A*(*t*) and *B*(*t*). It is easy to show that, under this condition, *k_+_
*is bigger than *k*_−_, so this intersection is unavoidable. The concentration at this intersection point *A*(*t**) = *B*(*t**) = 1/2. Consequently, in this case, *the time of intersection* of reactant–product dependencies is *the half-life time*, which is well known in chemical kinetics.

This fact does not depend on the details of the kinetic behavior, whether the system is reversible or irreversible, linear or non-linear, and, finally, ideal or non-ideal.

Also, the ’swapped equilibrium’ (SE) experiment can be performed when the initial concentrations of two chemical species are taken as their equilibrium concentrations but swapped, see [27]. In this experiment, another intersection of kinetic dependencies can be obtained as well.

### 3.1. Ideal System

The kinetic model is as follows:*A*′ = −*k_+_A + k*_−_(1 − *A*) = −(*k_+_ + k*_−_)*A + k*_−_.
Therefore, *A*(*t*) = *A_eq_* + (*A*_0_ − *A_eq_*)exp(λ*t*), *B*(*t*) = *B_eq_* + (*B*_0_ − *B_eq_*)exp(λ*t*), *A_eq_ = k*_−_/(*k_+_ + k*_−_) = 1/(*K_eq_ +* 1), *B_eq_ = k*_+_/(*k_+_ + k*_−_) = *K_eq_*/(*K_eq_* + 1), and λ = −(*k_+_ + k*_−_).

The linear time of relaxation is as follows:τ = 1/|λ|= 1/(*k_+_ + k*_−_).
The relaxation time for reaching the concentration *A*(*t*) when starting from the concentration *A*_0_ is as follows:*t* = *t*(*A*, *A*_0_) = τ ln((*A*_0_ − *A_eq_*)/(*A* − *A_eq_*)).
For the half-life point, (*A*_0_ − *A_eq_*)/(*A* − *A_eq_*) = 2 and *t*_1/2_ = τ ln2.

For the intersection point, *A*(*t*) = *B*(*t*) = ½. Therefore, the intersection point is the half-life-point as well. At *A***_0_** = 1, the time to reach the intersection point is as follows:*t_int_* = ln[2*K_eq_*/(*K_eq_ −* 1)]/(*k_+_ + k*_−_) = τ ln[2*K_eq_*/(*K_eq_ −* 1)].
In the ‘swapping equilibrium’ experiment, for the intersection point, *t_swap_*, at *A*_0_ = *B_eq_*,*t_swap_
*= ln(2)/(*k_+_ + k*_−_) = τ ln2.
Therefore, *t_swap_* coincides with *t*_1/2_.

Then, the two-time ratio,t_int_/t_swap_ = ln[2K_eq_/(K_eq_ − 1)]/ln2.

**Example 1.** 
*Consider a system with k_+_ = 2 s^−1^, k_−_ = 1 s^−1^, K_eq_ = 2, A_eq_ = 1/3, B_eq_ = 2/3, λ = −3 s^−1^, and τ = 1/3 s. For A_0_ = 1, the half-life is t = t_int_ = 0.46 s. In the “swapping equilibrium” experiment, A_0_ = B_eq_ and t_swap_ = 0.23 s. Therefore, the ratio is t_int_/t_swap_ = 2, illustrating the difference between these times.*


The main result of this section is the following: in this ideal system with the single reaction, there are some primary characteristics of general interest, i.e., the intersection time, *t_int_,* the ‘swap’ time, *t_swap_*, and the ratio, *t_int_/t_swap_*.

### 3.2. Non-Ideal System

In the Marcelin–de Donde (MDD) model, the reaction rates are expressed by the relationships *r_+_* = *k_+_*exp μ*_A_* and *r_−_ = k_−_*exp μ*_B_*, where *k_+_* and *k_−_* are kinetic coefficients; μ*_A_*= μ*_A_*_0_ + ln *A* + *g_A_*, μ*_B_*= μ*_B_*_0_ + ln *B* + *g_B_* are the potentials of the reactants; μ*_A_*_0_ and μ*_B_*_0_ are the initial potentials of the reactants; and *g_A_* = *g_A_*(*A*, *B*) and *g_B_* = *g_B_*(*A*, *B*) are the corrections for the non-idealities of the reactants. If the non-ideality corrections are linear with respect to the reactant concentrations, *g_A_
*= *a*_11_*A* + *a*_12_*B*; *g_B_
*= *a*_21_*A* + *a*_22_*B*, where *a*_11_, *a*_12_ and *a*_21_, *a*_22_ are the coefficients of the non-idealities of reactants A and B. Then, the kinetic model can be written as follows:*A*′ = −*k_+_A*exp(*a*_11_*A* + *a*_12_*B*) +* k*_−_*B*exp(*a*_21_*A* + *a*_22_*B*).
The reaction dynamics were investigated for different combinations of non-ideality coefficients considering ‘soft’ non-ideality with *a*_11_ = *a*_21_, *a*_12_ = *a*_22_. As mentioned earlier, this condition ensures the constancy of the equilibrium, meaning that the equilibrium composition does not change in comparison with the ideal system. The following cases are considered:


1.The non-ideality functions of the reactants are the same, with all individual non-ideality coefficients being equal, i.e., *a*_11_ = *a*_21_ = *a*_12_ = *a*_22_ ≡ *p* ≠ 0 (uniform non-ideality). In this case, λ* = − (*k*_+_ + *k*_−_)exp(*p*) and τ* = 1/[(*k*_+_ + *k*_−_)exp(*p*)] = τ exp(−*p*). Therefore, when *p* > 0, the linear relaxation time decreases by a factor of exp(*p*), and when *p* < 0, it increases by the same factor.


Accordingly,*t*_int_* = τ* ln[2*K_eq_*/(*K_eq_ −* 1)] = τ exp(−*p*) ln[2*K_eq_*/(*K_eq_ −* 1)].*t*_swap_
*= τ* ln2 = τ exp(−*p*) ln 2.*t**_*int*/_*t***_swap_* = ln[2*K**_eq_*/(*K**_eq_* − 1)]/ln2 = *t**_int/_**t**_swap_*.
Thus, for the uniform non-ideality, the ratio of the intersection time to the ‘swap’ time remains unchanged.

2.The non-ideality functions of the reactants are the same, but the individual non-ideality coefficients are different, i.e., *a*_11_ = *a*_21_ ≠ *a*_12_ = *a*_22_ (the non-uniform non-ideality coefficients). In this case,

τ* = τ exp(−*p**), where *p** ≡ *a*_11_*A_eq_* + *a*_12_*B_eq._*
Consequently, in this case, the characteristic *p** is a function of both the non-ideality coefficients and the equilibrium concentrations.

Here, τ* decreases when *p** > 0 and the reaction speeds up by a factor of exp(*p**), while when *p** < 0, τ∗ increases and the reaction slows down by the same factor.

The kinetic model becomes the following:*A*′ = (*k*_−_*B* − *k_+_A*)exp(*a*_11_*A* + *a*_12_*B*).
Integrating this equation over the intervals *t* ∈ [0, *t**], *A* ∈ (*A*_0_, *A**), and *B* ∈ (*B*_0_, *B**) gives the relaxation duration from the initial conditions of *A*_0_, *B*_0_ to the intersection point of *A**, *B** for non-uniform non-ideality, as follows:*t** = τ τ*[Ei(1, (*a*_12_ − *a*_11_)(*k_−_B** − *k*_+_*A**)τ) − Ei(1, (*a*_12_ − *a*_11_)(*k_−_B*_0_ − *k*_+_*A*_0_)τ)],
where Ei is the exponential integral [28], see our Appendix A. Note that this relationship allows for calculating the time of intersection of curves *A*(*t*) and *B*(*t*) only if *k_+_* > *k*_−_; otherwise, these curves do not intersect, meaning that point *A** does not exist.

For the intersection point with *A*_0_ = 1, *B*_0_ = 0,*t*_int_* = τ τ*[Ei(1, (*a*_12_ − *a*_11_)(*k_−_* − *k*_+_)τ/2) − Ei(1, (*a*_12_ − *a*_11_)(−*k*_+_)τ)].
In the ‘swapping equilibrium’ experiment, for the intersection point with *A*_0_ = *B_eq_*, *B*_0_ = *A_eq_*,*t*_swap_*= τ τ*[Ei(1, (*a*_12_ − *a*_11_)(*k_−_* − *k*_+_)τ/2) − Ei(1, (*a*_12_ − *a*_11_)(*k_−_B*_0_ − *k*_+_*B_eq_*)τ)].
The two-time ratio is a function of the non-ideality coefficients and initial conditions, as follows:*t*_int_/t*_swap_
*= Ei(1, (*a*_12_ − *a*_11_)(*k_−_* − *k*_+_)τ/2) − Ei(1, (*a*_12_ − *a*_11_)(− *k*_+_)τ)]/[Ei(1, (*a*_12_ − *a*_11_)(*k_−_* − *k*_+_)τ/2) − Ei(1, (*a*_12_ − *a*_11_)(*k_−_B*_0_ − *k*_+_*B_eq_*)τ)].
Thus, in the case of non-uniform non-ideality, the two-time ratio changes in comparison with the ideal system. This represents a fingerprint of the non-ideal system that can be detected experimentally.

**Example 2.** 
*Evolution of the times to reach the given concentration.*


Let us analyze some non-ideal cases at *K_eq_
*= 2, *A_eq_
*= 1/3, *B_eq_
*= 2/3, and *A* = 1/2, as follows: (1) *a*_11_ = 0 and *a*_12_ = 0 (‘zeroth’ non-ideality); (2) *a*_11_ = ¼ and *a*_12_ = −1/2 (‘weak’ non-ideality); (3) *a*_11_ = ½ and *a*_12_ = −1 (non-ideality is ‘stronger’,); and (4) *a*_11_ = 1 and *a*_12_ = −2 (‘strong’ non-ideality). For these cases, the following dependences for *t** from ln((*A*_0_ − *A_eq_*)/(*A* − *A_eq_*)) are shown in Figure 2.

For the ideal system, the relaxation time depends linearly on the logarithm of the scaled initial conditions, while for the non-ideal system, a similar dependence becomes non-linear. The bigger the non-ideality coefficients, the stronger is the apparent non-linearity.

**Example 3.** *Rapid test. The evolution of the two-time-ratio is shown in Table 1*.

In the table, each of the two-time ratios *t*_int_/t*_swap_* represents a rapid, single-point test that can be used for identifying non-ideality. The more the experimental ‘swap’ value *t*_int_*/*t*_swap_* deviates from the ideal value of 2.0, the more likely it is that the system is non-ideal.

As can be seen from these examples, for the non-ideal system, the two-time ratio, i.e., the ratio of the intersection time to the ‘swap’ time, differs from the same ratio for the ideal system. Such a difference can be used as a fingerprint for distinguishing these systems. As mentioned, in all cases, the equilibrium concentrations for the ideal and corresponding non-ideal system are identical. We emphasize that, for approximating the real conditions, corrections for the non-ideality were taken to be moderate, i.e., about 20%.

## 4. Linear Two-Step Mechanism

The detailed mechanism analyzed is the following consecutive scheme:A ↔ B ↔ C,(6)
where A, B, and C—reactants. For this mechanism, the kinetic model (2) is represented as follows:*A*′ = −*r*_1_, *B*′ = *r*_1_ *− r*_2_, *C*′ = *r*_2_, *A*(0) = *A*_0_, *B*(0) = *B*_0_, *C*(0) = *C*_0_,(7)
where *A* = *A*(*t*), *B* = *B*(*t*), and *C* = *C*(*t*)—reactant concentrations and *r*_1_ and *r*_2_—the rates of the first and second steps, respectively. In this system, the conservation law *A*(*t*) +* B*(*t*) +* C*(*t*) = 1 is satisfied. For this system, the phenomenon of conservatively perturbed equilibrium (CPE), described in the introduction, will be compared for the ideal and non-ideal cases. The ideal system behavior was studied in detail in [15].

A special kinetic event for this system is the appearance of the over-equilibrium, i.e., the concentration extremum, which is observed if the kinetic reactant dependence starts from its equilibrium concentration. Some other concentrations remain as the equilibrium ones.

As mentioned, this experiment is performed at the same total amount of each chemical element and a fixed temperature.

For the closed system, this over-equilibrium (extremum) is unavoidable.

### 4.1. Ideal System

In an ideal system, the kinetic model can be represented as follows:*A*′ = −*k_+_*_1_*A* + *k*_−1_*B*, *C*′ = *k_+_*_2_*B* − *k*_−2_*C*, *B* = 1 − *A* − *C*.
The equilibrium composition is determined by the expressions *A_eq_* = *k*_−1_*k*_−2_/Δ, *B_eq_* = *k*_+1_*k*_−2_/Δ, *C_eq_* = *k*_+1_*k*_+2_/Δ, and Δ ≡ *k*_+1_*k*_+2_ +* k*_+1_*k*_−2_+* k*_−1_*k*_−2_. The equilibrium constants take the values *K*_1_ = *B_eq_*/*A_eq_* and *K*_2_ = *C_eq_/B_eq_*. The linear relaxation time is calculated as follows τ = 1/min|Re(λ_1_, λ_2_)| = 2/|σ + *D*|, where the eigenvalues λ_1_, λ_2_ = (σ ± *D*)/2, σ = −(*k*_1_ +* k*_−1_+* k*_2_ +* k*_−2_), *D* = (σ^2^ − 4Δ)^1/2^, and the equilibrium is stable.

In this two-step mechanism, under ideal assumptions, the time of CPE depends only on the eigenvalues which are functions of kinetic coefficients *t* = ln(λ_1_/λ_2_)/(λ_2_ − λ_1_). Detailed computer experiments demonstrated that this time does not depend on the reactant concentrations [15]. Consequently, it is a kinetic fingerprint of the ideal system.

**Example 4.** 
*Let us set the rate coefficients arbitrarily k_+1_ = 4; k_−1_ = 3; k_+2_ = 2; and k_−2_ = 1, then, with the ideal kinetics, the equilibrium constants of the stages are K_1_ = 1.33 and K_2_ = 2.00, the equilibrium composition is A_eq_ ≈ 0.20, B_eq_ ≈ 0.27, C_eq_ ≈ 0.53, σ = −10, D = 40, Δ = 15, λ_1_, λ_2_ = (−8.16, −1.83), and the linear relaxation time is τ ≈ 0.54. The time of appearance of CPE is t* ≈ 0.24 and does not depend on the initial concentrations of the reactants.*


### 4.2. Non-Ideal System

The chemical potentials (5) of the reactants of this system are presented as follows:μ*_A_* = μ^0^*_A_* + ln *A + g_A_*, μ*_B_* = μ^0^*_B_* + ln *B + g_B_*,μ*_C_* = μ^0^*_C_* + ln *C + g_C_*,(8)
where *g_A_* (*A*, *B*, *C*), *g_B_* (*A*, *B*, *C*), and *g_C_* (*A*, *B*, *C*)—correction functions for the non-ideal reactants.

The thermodynamic conditions for these potentials are the following:*∂g_B_*/*∂A *= *∂g_A_*/*∂B*, *∂g_A_*/*∂C *= *∂g_C_*/*∂A*, *∂g_B_*/*∂C *= *∂g_C_*/*∂B*,(9)*∂g_A_*/*∂A* ≥ 0, (*∂g_A_*/*∂A*) (*∂g_B_*/*∂B*) ≥ (*∂g_A_*/*∂B*)^2^, det(***M***) ≥ 0.
The rates of steps (4) for the potentials (8) are the following:*r*_1_ = *r*_1_^0^[exp(μ^0^*_A_*) *A* exp(*g_A_*) − exp(μ^0^*_B_*) *B* exp(*g_B_*)],(10)*r*_2_ = *r*_2_^0^[exp(μ^0^*_B_*) *B* exp(*g_B_*) − exp(μ^0^*_C_*) *C* exp(*g_C_*)].
Considering (7)–(10), the kinetic model of the non-ideal system can be written as follows:*A*′ = −*k_+_*_1_*A*exp(*g_A_*) + *k*_−1_*B*exp(*g_B_*), *C*′ = *k_+_*_2_*B*exp(*g_B_*) − *k*_−2_*C*exp(*g_C_*),(11)
where *B* = 1 − *A − C*, *k_+_*_1_ = *k*^0^*_+_*_1_exp(μ^0^*_A_*), *k*_−1_ = *k*^0^*_−_*_1_exp(μ^0^*_B_*), *k_+_*_2_ = *k*^0^*_+_*_2_exp(μ^0^*_B_*), *k*_−2_ = *k*^0^*_−_*_2_exp(μ^0^*_C_*) and *k*^0^*_+_*_1_, *k*^0^*_−_*_1_, *k*^0^*_+_*_2_, *k*^0^*_−_*_2_—pre-exponential coefficients. Under the equilibrium conditions, the rates of all steps are zero, and*k_+_*_1_*A*exp (*g_A_*) = *k*_−1_*B*exp(*g_B_*), *k_+_*_2_*B*exp(*g_B_*) = *k*_−2_*C*exp(*g_C_*).(12)
The ‘non-ideal’ equilibrium constants of the steps are determined as follows:*K**_1_ = *B/A* = (*k_+_*_1_/*k*_−1_) exp(*g_A_ − g_B_*) = *K*_1_exp(*g_A_ − g_B_*),(13)*K**_2_ = *C/B* = (*k_+_*_2_/*k*_−2_) exp(*g_B_ − g_C_*) = *K*_2_ exp(*g_B_ − g_C_*),
where *K*_1_ and *K*_2_ are the “ideal” equilibrium constants.

In accordance with our requirement (‘soft’ non-ideality), the equilibrium composition of the non-ideal system does not change compared with the ideal one.

It follows from (12) and (13) that this requirement is fulfilled if, and only if, the correction functions for the non-ideality of the reactants are equal, as follows:*g_A_* = *g_B_* = *g_C_*.(14)

If these requirements are met, the ‘non-ideal’ equilibrium constants and the equilibrium composition necessarily coincide with the ideal ones, respectively, as follows:*K**_1_ = *K*_1_ = *B_eq_*/*A_eq_*, *K**_2_ = *K*_2_ = *C_eq_/B_eq_*,(15)

Relationships (15) define the conditions of ‘soft’ non-ideality for the analyzed system. If the conditions (14) are not met, the equilibrium composition is obviously changed.

Let us detail criterion (14) for linear non-ideal correction functions. These functions regarding the reactant concentrations are assumed as follows:*g_A_* = *a*_11_*A* + *a*_12_*B* + *a*_13_*C*,*g_B_* = *a*_21_*A* + *a*_22_*B* + *a*_23_*C*,(16)*g_C_* = *a*_31_*A* + *a*_32_*B* + *a*_33_*C*,
where *a*_11_, *a*_12_, *a*_13_, *a*_21_, *a*_22_, *a*_23_, *a*_31_, *a*_32_, and *a*_33_—coefficients of the non-ideality. Then, the potentials (8) are the following:μ*_A_* = ln*A* + *a*_11_*A* + *a*_1_μ*_B_* = ln*B + a*_21_*A* + *a*_22_*B* + *a*_23_*C*, (17)μ*_C_* = ln*C + a*_31_*A* + *a*_32_*B* + *a*_33_*C.*(18)
Considering (16)–(18), a kinetic model (11) is presented as follows:*A*′ = −*k_+_*_1_ *A* exp (*a*_11_*A* + *a*_12_*B* + *a*_13_*C*) + *k*_−1_*B*exp (*a*_21_*A* + *a*_22_*B* + *a*_23_*C*),(19)*C*′ = *k_+_*_2_ *B* exp (*a*_21_*A* + *a*_22_*B* + *a*_23_*C*) − *k*_−2_ *C* exp (*a*_31_*A* + *a*_32_*B* + *a*_33_*C*).
For the non-ideal equilibrium (12), equations will be written as follows:*k_+_*_1_*A*exp(*a*_11_*A* + *a*_12_*B* + *a*_13_*C*) = *k*_−1_*B*exp(*a*_21_*A* + *a*_22_*B* + *a*_23_*C*),(20)*k_+_*_2_*B*exp(*a*_21_*A* + *a*_22_*B* + *a*_23_*C*) = *k*_−2_*C*exp(*a*_31_*A* + *a*_32_*B* + *a*_33_*C*).
Consequently, Equation (13) will be written as follows:*K**_1_ ≡ *B/A* = *K*_1_exp[(*a*_11_
*− a*_21_)*A* + (*a*_12_
*− a*_22_)*B* + (*a*_13_
*− a*_23_)*C*)],(21)*K**_2_ ≡ *C/B* = *K*_2_exp[(*a*_21_
*− a*_31_)*A* + (*a*_22_
*− a*_32_)*B* + (*a*_23_
*− a*_33_)*C*)],
where the exponential factors express the non-ideality coefficients of the equilibrium constants.

*Introducing the ‘soft’ non-ideality*. From (14)–(21), it follows that, for the linear non-ideality functions, the equilibrium constants of the steps do not change if, and only if, the equalities are satisfied (if only one of them is satisfied, then only one corresponding equilibrium constant is preserved).*a*_11_*A* + *a*_12_*B* + *a*_13_*C* = *a*_21_*A* + *a*_22_*B* + *a*_23_*C*,(22)*a*_21_*A* + *a*_22_*B* + *a*_23_*C* = *a*_31_*A* + *a*_32_*B* + *a*_33_*C*,
where *a*_11_, *a*_21_, and *a*_31_—non-ideality coefficients related to the reactant A. Similarly, the coefficients *a*_12_, *a*_22_, and *a*_32_ are related to the reactant B, and the coefficients *a*_13_, *a*_23_, and *a*_33_ are related to reactant C. These three groups of coefficients can be named as the individual non-ideality coefficients, which correspond to the chosen substances A, B, and C, respectively. So, the ‘soft’ non-ideality is defined by Equation (22).

The following conditions*a*_11_ = *a*_21_ = *a*_31_, *a*_12_ = *a*_22_ = *a*_32_, *a*_13_ = *a*_23_ = *a*_33_(23)
can be interpreted as sufficient conditions of ‘soft’ non-ideality.

The thermodynamic conditions (9) for the potentials (18) can be presented as follows:*a*_12_ = *a*_21_, *a*_13_ = *a*_31_, *a*_23_ = *a*_32,_(24)*a* ≥ 0, *ab* − *a*_12_*a*_21_ ≥ 0, *abc* + *a*_12_*a*_23_*a*_31_ − *ba*_13_*a*_31_ − *ca*_21_*a*_12_ − *aa*_23_*a*_32_ ≥ 0, (25)
where *a* ≡ 1/*A* + *a*_11_, *b* ≡ 1/*B* + *a*_22_, and *c* ≡ 1/*C* + *a*_33_.

It should be mentioned that the non-ideality coefficients *a*_11_, *a*_12_, and *a*_13_ can be different. Assuming the equality of symmetric coefficients (24) and comparing them with (23), is easy to show that individual non-ideality coefficients are equal. In the general case, criterion (22) can be satisfied even if conditions (23) are violated.

For the non-ideal system, all or some non-ideality coefficients are nonzero. The linear relaxation times are determined by the following expressions, respectively:τ* = 2/|σ* + *D**|,(26)
where σ* and *D** depend on the non-ideality coefficients and equilibrium concentrations of the reactants, as follows:σ* = −{[1 + *A*(*a*_12_ − *a*_11_)]*k*_1_*p*_1_ − [*k*_−1_ + *k*_2_ + *k*_−1_*B(a*_22_ − *a*_21_) + *k*_2_*B*(*a*_22_ − *a*_23_)]*p*_2_ − [1 − *C*(*a*_32_ − *a*_33_)]*k*_−2_*p*_3_}, (27)
where *p*_1_ = exp(*a*_11_*A* + *a*_12_*B* + *a*_13_*C*), *p*_2_ = exp(*a*_21_*A* + *a*_22_*B* + *a*_23_*C*), and *p*_3_ = exp(*a*_31_*A* + *a*_32_*B* + *a*_33_*C*).

Obviously, these characteristics can change the sign. The expression for *D** is not presented, since it is too big for this paper. However, readers can easily derive this equation using the computer algebra methods. Consequently, in the general case, i.e., for arbitrary non-ideality coefficients, the relaxation process may exhibit quite complicated behavior, say relaxation with damped oscillations.

In a particular case, with equal non-ideality coefficients (*a_ij_* = *p*), σ* and *D** are expressed by the following simple relationships: σ* = σ exp(*p*) < 0, *D** = *D* exp(*p*) > 0, respectively. Therefore, λ*_1_, λ*_2_ < 0, i.e., the system equilibrium remains stable. From (25) and (26), the following can be observed:τ*/τ =|σ + *D*|/|σ* + *D**| = exp(−*p*), (28)
where τ and τ* are linear relaxation times for the ideal and non-ideal systems, respectively.

Considering ‘soft’ non-ideality, we will investigate the influence of the non-ideality coefficients and initial conditions for mechanism (6) with the potentials (18) on the time of the CPE-extremum and its parameters.

## 5. Different Scenarios of Non-Ideal Dynamics

1.All coefficients of non-ideality are equal: *a_ij_* = *p* (the ‘equal’ non-ideality). Then, the ‘soft’ non-ideality conditions (22) and (23) are satisfied for any *p*, and the thermodynamic conditions (24) and (25) are valid only for *p* ≥ 0. Hence, based on (28), τ* = τ exp(−*p*), the reaction should accelerate by a factor of e*^p^* when *p* ≥ 0 and decelerate when *p* ≤ 0. Consequently, the observation time of the CPE effect should also shift by a factor of e*^p^*.2.Not all non-ideality coefficients are equal (the ‘nonequal’ non-ideality). In this more general case, the conditions of ‘soft’ non-ideality (22) and (23) and the thermodynamic conditions (24) and (25) may be violated. The equilibrium composition is maintained only if the conditions of the ‘soft’ non-ideality (22) are satisfied. In addition, from (27) and (28), it follows that the linear relaxation time τ* depends on the values of all non-ideality coefficients. Accordingly, the moment of observation of the CPE effect should also shift.3.Non-ideality coefficients may have any value (the ‘arbitrary’ non-ideality). In this most general case, conditions (22)–(25) are certainly not satisfied. Equilibrium and nonequilibrium characteristics can differ significantly from the characteristics of ‘equal’ and ‘non-equal’ non-ideality. As shown above, regarding the note after (27), this case can lead to non-physical results and, therefore, is not considered here.

## 6. Computer Calculations

In our calculations, all scenarios mentioned will be analyzed regarding the features of CPE and comparing the non-ideal cases with the ideal ones. Let assume that *k_+_*_1_ = 4; *k*_−1_ = 3; *k_+_*_2_ = 2; and *k*_−2_ = 1 and *K*_1_ = 1.33, *K*_2_ = 2.00, *A_eq_* ≈ 0.20, *B_eq_* ≈ 0.27, and *C_eq_* ≈ 0.53, respectively. In an ideal system, τ = 0.54, τ_ε_ ≈ 2.50, and the CPE time is *t** ≈ 0.24.

Case 1. The non-ideality coefficients are equal and non-zeroth, as shown in Figure 3.

In this case, the non-ideal equilibrium constants are *K**_1_ = 1.33 = *K*_1_ and *K**_2_ = 2.00 = *K*_2_, respectively. Comparing the ‘non-ideal’ equilibrium composition with the ‘ideal’ one, it remains the same, *A*_eq_* ≈ 0.20 = *A_eq_*, *B*_eq_* ≈ 0.27 = *B_eq_*, and *C*_eq_* ≈ 0.53 = *C_eq_*.

In Figure 3a,b, it is shown that, considering equal positive coefficients of non-ideality *p* = 1, the concentration of reactant A goes through the extremum at the temporal point *t**, which is observed earlier than the corresponding one for the ideal system. Therefore, the CPE effect is shifted “to the left”.

From (26)–(28), it follows that σ* = −27.18, *D** = 108.73, λ*_1_, λ*_2_ = (−22.19, −4.99); τ ≈ 0.20, τ_ε_ ≈ 0.90, which means that the relaxation times decrease by a factor of e*^p^* = e; τ* from 0.54 to 0.20 and τ*_ε_ from 2.50 to 0.90, respectively. The appearance time of the CPE *t** ≈ 0.09 also decreases accordingly, but does not depend on the initial concentrations of the reactants

The computer calculations demonstrated that, if *A*_0_ = *A_eq_*, the reactant concentrations at the CPE point *t** are *A**(*t**) ≈ 0.14, *B*(*t**) ≈ 0.18, and *C*(*t**) ≈ 0.68. Then, the ratio *B*(*t**)*/A**(*t**) ≈ 1.35 ≈ *K**_1_ = 1.33 is approximately equal to the equilibrium constant of the first step. An error is (1.35 − 1.33)/1.35 ≈ 0.01%. Hence, the CPE point of component A is the momentary equilibrium of the first step. This is understandable, because component A participates in the single reaction. As for the ratio *C*(*t**)*/B*(*t**) ≈ 3.67, it is not equal to the equilibrium constant of the second step *K**_2_. The error is (3.67 − 2.00)/3.67 = 45.57%. In contrast to component A, component B participates in two reactions, and its extremum cannot be considered as the momentary equilibrium of the single reaction.

If *B*_0_ = *B_eq_*, at the CPE point, *A*(*t**) ≈ 0.45, *B**(*t**) ≈ 0.39, and *C*(*t**) ≈ 0.16. The ratios *B**(*t**)*/A*(*t**) ≈ 0.88 and *C*(*t**)*/B**(*t**) ≈ 0.40 are not equal to the equilibrium constants of the first and second step, *K**_1_ and *K**_2_, respectively, e.g., the error for the second step is (2.00 − 0.40)/2 = 79.98%. As mentioned, this is caused by the fact that *B* participates in two steps, not in the single one.

Analogously, if *C*_0_ = *C_eq_*, at the corresponding CPE point, *A*(*t**) ≈ 0.26, *B*(*t**) ≈ 0.25, and *C**(*t**) ≈ 0.49 and the ratio *B*(*t**)*/A*(*t**) ≈ 0.95 is not equal to *K**_1_. The error is (1.33 − 0.95)/1.33 = 28.62%. However, in this case, the ratio *C**(*t**)*/B*(*t**) ≈ 1.98 is equal to the equilibrium constant of the second step *K**_2_. The error is (2.00 − 1.98)/2 ≈ 0.01%. Therefore, based on the CPE experiment, it can be concluded that component C participates in the single reaction, i.e., in the second one.

In Figure 3c,d, it is shown that, under assumptions on the equal but negative coefficients of non-ideality *p* = −1, the CPE effect causes a shift ‘to the right’. The relaxation times increase by e times: τ* from 0.54 to 1.48 and τ*_ε_ from 2.5 to 6.80, respectively.

The time of appearance of CPE, *t** ≈ 0.64, increases accordingly. It also does not depend on the initial concentrations of the reactants.

Case 2. Let us assume that the non-ideality coefficients are different, but satisfy the criterion of ‘soft’ non-ideality. The dynamics are shown in Figure 4.

In this case, as in the previous ones, the non-ideal equilibrium constants and the non-ideal equilibrium composition do not change compared with the ideal ones.

In Figure 4a,b, it is demonstrated that at ‘non-uniform’ positive non-ideality coefficients, the concentrations of the reactants exhibit an extremum (minimum) earlier than the corresponding concentrations in an ideal system.

From (26)–(28), it follows that σ* = −103.12, *D** = 65.22, (λ*_1_, λ*_2_) = (−84.17, −18.95), τ ≈ 0.05, τ_ε_ ≈ 0.25. This means that the relaxation times significantly decrease compared to ‘equal’ non-ideality: τ* from 0.54 to 0.05 and τ*_ε_ from 2.50 to 0.25, respectively. The time of appearance of CPE, *t** also decreases accordingly, but now depends on the initial concentrations of the reactants, see Figure 4a. It is different from the ideal system.

At *A*_0_ = *A_eq_*, the concentrations of the reactants at the point *t** are *A**(*t**) ≈ 0.14, *B*(*t**) ≈ 0.19, and *C*(*t**) ≈ 0.67. In this case, the ratio *B/A** ≈ 1.35 is equal to the equilibrium constant of the first step *K**_1_, with an error of (1.35 − 1.33)/1.35 ≈ 1.46%. As for the ratio *C/B* ≈ 3.53, it is not equal to the equilibrium constant of the second step *K**_2_. The error is about (3.53 − 2.00)/2.00 ≈ 76.50%.

At *B*_0_ = *B_eq_*, the concentrations of the reactants at point *t** are *A*(*t**) ≈ 0.58, *B**(*t**) ≈ 0.40, and *C*(*t**) ≈ 0.02. According to this experiment, the ratio *B**/*A* ≈ 0.69 is not equal to the equilibrium constant of the first step *K**_1_. The error is (1.33 − 0.69)/1.33 ≈ 48.12%, and the ratio *C*/*B** ≈ 0.14 is not equal to the equilibrium constant of the second step *K**_2_. The error is (2.00 − 0.14)/2.00 ≈ 93.00%.

Figure 4c,d show that at ‘non-uniform’ negative non-ideality coefficients, the concentrations of the reactants pass through an extremum (maximum) later than the concentrations of the reactants of the ideal system. The relaxation times are increased significantly compared to the ‘equal’ non-ideality and, according to (26)–(28), are τ* ≈ 5.25 and τ*_ε_ ≈ 20.0.

The time of CPE extremum *t*^∗^ is increased accordingly, becoming dependent on the initial reactant concentrations. It is different from the behavior of corresponding ideal system, where the time of the CPE-extremum does not depend on these concentrations. Such a difference can be considered as a fingerprint of non-ideal behavior, see Figure 5a,b.

The CPE extremum significantly depends on the initial conditions. It can be considered as the fingerprint of the non-ideality of this system.

## 7. Conclusions

For the closed gradientless reactor, two cases of reversible first-order reactions were analyzed, i.e., the single-step reaction and the two-step consecutive reaction. In both cases, two models were investigated, the ideal one and non-ideal one (Marcelin–de Donder).

For the single-step ideal and non-ideal models, the behavior of the intersection point time was chosen as the fingerprint of the non-ideality. For the non-ideal model, it depended on the initial conditions.

For the ideal two-step model, the Conservatively Perturbed Extremum (CPE)-extremum time was well-defined. For the non-ideal two-step model, the thermodynamic restrictions and linear corrections for non-ideality were considered under the assumption of ‘soft’ non-ideality. For this model, the CPE effect was observed for any coefficients of non-ideality, equal and nonequal. For both ideal and non-ideal systems, the equilibrium composition remained the same, however, the relaxation times changed.

When some nonideality coefficients were not equal, the time of the appearance of the CPE point (over-equilibrium) depended on the initial concentrations significantly. This case can be considered as the more general one. The obtained difference in the CPE-extremum time between the ideal and non-ideal systems was a proposed kinetic fingerprint for distinguishing these systems

Positive non-ideality coefficients always sped up the relaxation, while negative coefficients slowed it down. The CPE-extremum time decreased or increased, respectively, compared to the ideal system. With equal coefficients of non-ideality, the chemical system behaved like an ideal one. This case can be considered rather as an exception.

## Figures and Tables

**Figure 1 entropy-27-00077-f001:**
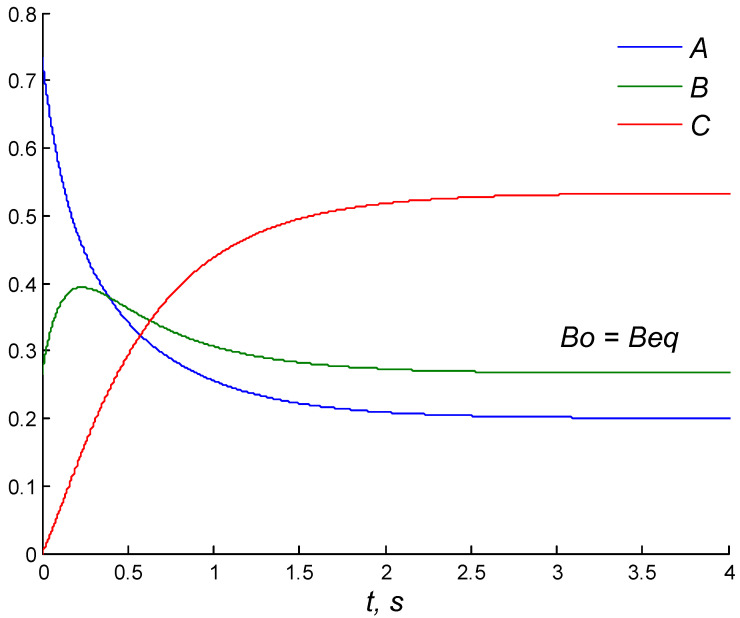
CPE effect in ideal two-step mechanism A ↔ B ↔ C. Dynamics of reactant concentrations at *A*(*t*) +* B*(*t*) +* C*(*t*) = 1 (conservation law); *B*_0_, *B_eq_*—initial and equilibrium concentrations of the reagent *B*.

**Figure 2 entropy-27-00077-f002:**
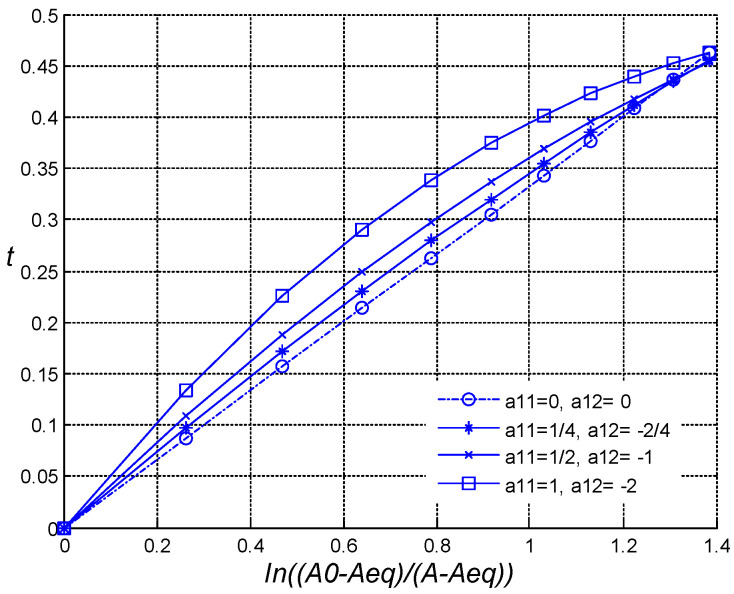
Relaxation time *t* from ln((*A*_0_ − *A_eq_*)/(*A* − *A_eq_*)) for different cases of the non-ideality: o—‘zeroth’; *—‘weak’; ×—non-ideality is ‘stronger’; and ☐—‘strong’.

**Figure 3 entropy-27-00077-f003:**
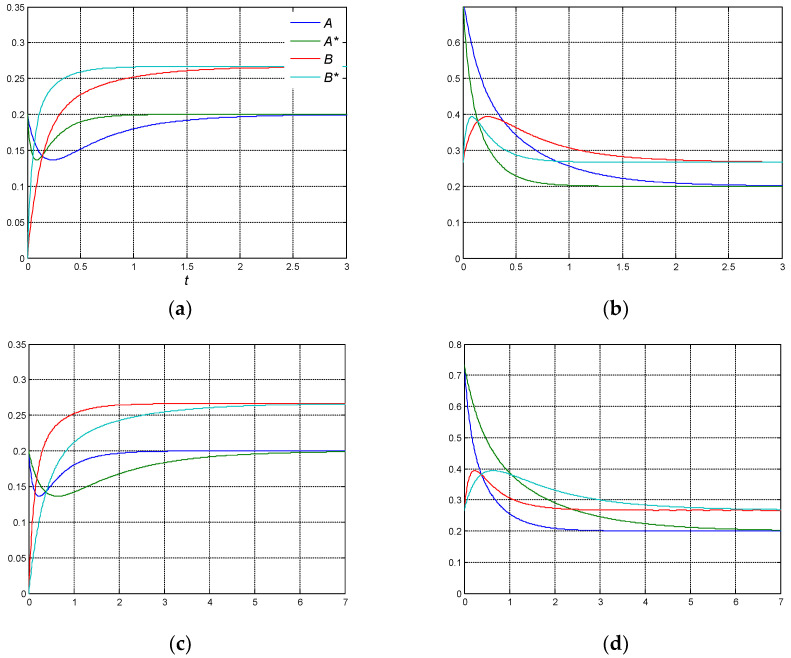
The ‘uniform’ non-ideality. Evolution of concentration of ideal (*A*, *B*) and non-ideal (*A**, *B**) components, respectively, at *a_ij_* = *p*: (**a**) *A*_0_ = *A_eq_*; *p = +*1; (**b**) *B*_0_ = *B_eq_*; *p = +*1; (**c**) *A*_0_ = *A_eq_*; *p = −*1; and (**d**) *B*_0_ = *B_eq_*; *p = −*1.

**Figure 4 entropy-27-00077-f004:**
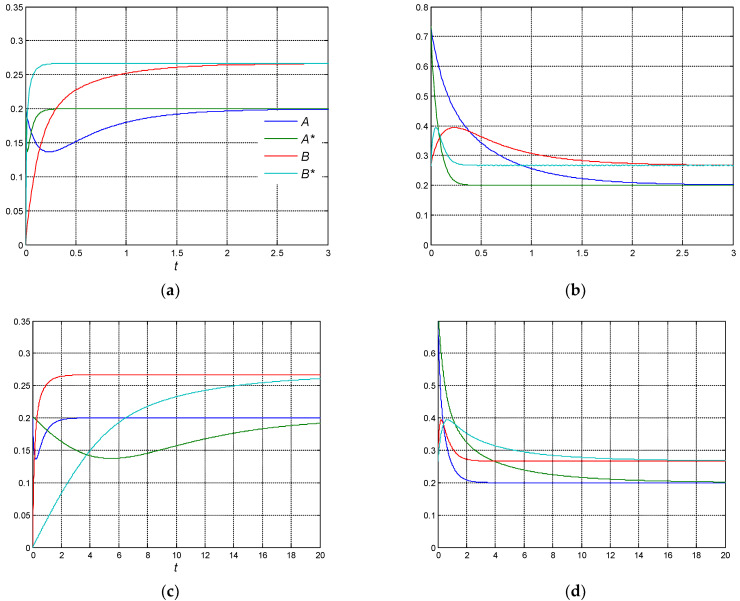
The ‘non-uniform’ non-ideality. Kinetic dependencies for ideal and non-ideal cases at *a*_11_ = *a*_21_ = *a*_31_ = 1, *a*_12_ = *a*_22_ = *a*_32_ = 2, *a*_13_ = *a*_23_ = *a*_33_ = 3: (**a**) *A*_0_ = *A_eq_*; (**b**) *B*_0_ = *B_eq_* and at *a*_11_ = −2, *a*_12_ = 1, *a*_13_ = −4, *a*_21_ = −2, *a*_22_ = 1, *a*_23_ = −4, *a*_31_ = −2, *a*_32_ = 1, *a*_33_ = −4; (**c**) *A*_0_ = *A_eq_*; and (**d**) *B*_0_ = *B_eq._*

**Figure 5 entropy-27-00077-f005:**
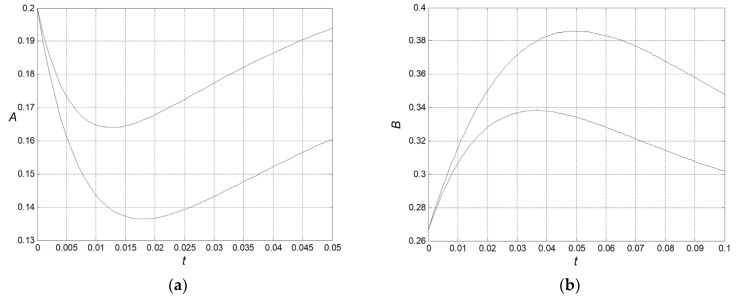
‘Non-uniform’ non-ideality. The time of CPE extremum, *t**, at different initial conditions: *A*_0_ = *A_eq_*, *B*_0_ = 0 (lower curve) and *A*_0_ = *A_eq_*, *B*_0_ = 0.1 (upper curve): (**a**) *a*_11_ = 1, *a*_12_ = 2, *a*_13_ = 3, *a*_21_ = 1, *a*_22_ = 2, *a*_23_ = 3, *a*_31_ = 1, *a*_32_ = 2, *a*_33_ = 3, *t** ≈ 0.018 → 0.013; and (**b**) *a*_11_ = 1/4, *a*_12_ = 2/4, *a*_13_ = 3/4, *a*_21_ = 1/4, *a*_22_ = 2/4, *a*_23_ = 3/4, *a*_31_ = 1/4, *a*_32_ = 2/4, *a*_33_ = 3/4, *t** ≈ 0.050 → 0.033.

**Table 1 entropy-27-00077-t001:** Evolution of the two-time ratio.

Coefficients of the Non-Ideality	*t*_int_*	*t*_swap_*	*t*_int_*/*t*_swap_*
*a*_11_ = 0, *a*_12_ = 0	0.46	0.23	2.0
*a*_11_ = 1/4, *a*_12_ = −1/2	0.4553	0.2479	1.8366
*a*_11_ = 1/2, *a*_12_ = −1	0.4533	0.2663	1.7025
*a*_11_ = 1, *a*_12_ = −2	0.4629	0.3084	1.5008
*a*_11_ = 1/10, *a*_12_ = −1/10	0.4587	0.2376	1.9307
*a*_11_ = 1/4, *a*_12_ = 2/4	0.3336	0.1618	2.0622
*a*_11_ = 5/4, *a*_12_ = 10/4	0.0918	0.0389	2.3580
*a*_11_ = −1/4, *a*_12_ = ½	0.5791	0.2489	2.33

## Data Availability

Data is contained within the article.

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
