# Peer review of "Distinguishing Ideal and Non-Ideal Chemical Systems Based on Kinetic Behavior"

_entropy, 2025, doi:10.3390/e27010077_

Round 1

Reviewer 1 Report

Comments and Suggestions for Authors

This paper focuses on differentiating between ideal and non-ideal chemical systems based on their kinetic behavior within a closed isothermal chemical environment. Non-ideality is examined using the non-ideal Marcelin-de Donde model. The theory and calculations of this paper are solid and the reviewer suggest it to be published after addressing the motivation in the introduction section.

Introduction section, ‘As for the non-ideal chemical isothermal system…’ the authors don’t explain well what are the differences between the non-ideal system and ideal system. The motivation is not clear.

Page 3 ‘where aij и bij - stoichiometric coefficients’ what does ‘и’ mean?

Author Response

Response to Reviewer 1

Comments 1. This paper focuses on differentiating between ideal and non-ideal chemical systems based on their kinetic behavior within a closed isothermal chemical environment. Non-ideality is examined using the non-ideal Marcelin-de Donde model. The theory and calculations of this paper are solid and the reviewer suggest it to be published after addressing the motivation in the introduction section.

Introduction section, ‘As for the non-ideal chemical isothermal system…’ the authors don’t explain well what are the differences between the non-ideal system and ideal system. The motivation is not clear.

Response 1. We decided to re-write some fragments of Section 2 “Kinetic Model” adding two sentences.

Instead of

“In the ideal kinetics of MAL, the rate of a reversible isothermal reaction [1, 12, 17-20] is equal to ri = r+i - r-i, where r+i = k+i ПAj aij, r-i = k-i ПAjbij  - the rate of forward and reverse reactions, respectively, 1/s.

In the non-ideal kinetics of MDD which is a generalization of the ideal kinetics of MAL, the rate of any step of a reversible isothermal reaction [10-14, 18-20] is :

ri = ri0[exp(åj aijmj) - exp(åj bijmj)], i = 1, 2, …,                                    (4)

where ri0 > 0 - the rate, forward or reverse, under the equilibrium conditions, mol/(l s); µj - the pseudo-chemical potential (hereinafter referred to as the potential) of the reactant Aj, dimensionless:

mj = m0j + ln Aj  + gj, j = 1, 2, …,              (5)

where m0j > 0 - constants; gj - correction terms for the non-ideality of reactants (gj = 0 for the ideal system); µj = µ*j/RT, µ*j - the chemical potential of the reactant Aj, J/mol; R is the universal gas constant, J/(mol oK); T - temperature, oK. “

the new text is written

“In our paper, the difference the ideal-system and non-ideal one is expressed in terms of chemical potentials.

In the ideal kinetics of MAL, the rate of a reversible isothermal reaction [1, 12, 17-20] is equal to ri = r+i - r-i, where r+i = k+i ПAj aij, r-i = k-i ПAjbij  - the rate of forward and reverse reactions, respectively, 1/s.

In the non-ideal kinetics of MDD which is a generalization of the ideal kinetics of MAL, the rate of any step of a reversible isothermal reaction [10-14, 18-20] is :

ri = ri0[exp(åj aijmj) - exp(åj bijmj)], i = 1, 2, …,                     (4)

where ri0 > 0 - the rate, forward or reverse, under the equilibrium conditions, mol/(l s); µj - the pseudo-chemical potential (hereinafter referred to as the potential) of the reactant Aj, dimensionless:

mj = m0j + ln Aj  + gj, j = 1, 2, …,                                                             (5)

where m0j > 0 - constants; gj - correction terms for the non-ideality of reactants (gj = 0 for the ideal system); µj = µ*j/RT, µ*j - the chemical potential of the reactant Aj, J/mol; R is the universal gas constant, J/(mol oK); T - temperature, oK.

A goal of this paper is to find the difference between the ideal and non-ideal case based on kinetic behavior”

Added three links to original works by Marcelin and De-Donder:

  1. De Donder, Th.; Van Rysselberghe, P. Thermodynamic theory of affinity. London: Oxford Univ. Press, 1936. 136 p.
  2. Van Rysselberghe, P. Reaction rates and affinities. J. Chem. Phys. 1958, 29, 640-642.
  3. Feinberg, M. On chemical kinetics of a certain class. Arch. for Rational Mech. and Analysis 1972, 46, 1-41.

Comments 2. Page 3 ‘where aij и bij - stoichiometric coefficients’ what does ‘и’ mean?

Response 2. Thank you for this remark. We agree with it. It is a typo.  On page 3 after formula (1) symbol ‘и’ is replaced with ‘and’.

Comments 1. This paper focuses on differentiating between ideal and non-ideal chemical systems based on their kinetic behavior within a closed isothermal chemical environment. Non-ideality is examined using the non-ideal Marcelin-de Donde model. The theory and calculations of this paper are solid and the reviewer suggest it to be published after addressing the motivation in the introduction section.

Introduction section, ‘As for the non-ideal chemical isothermal system…’ the authors don’t explain well what are the differences between the non-ideal system and ideal system. The motivation is not clear.

Response 1. Thank you for this comment. We agree with it. In Introduction, Paragraph 3, sentence 1 has been clarified as follows ‘As for the non-ideal chemical isothermal system, in which the MAL is violated, it was proposed the generalized Marcelin-de Donder (MDD) model in terms of the non-ideal chemical potentials.’

Comments 2. Page 3 ‘where aij и bij - stoichiometric coefficients’ what does ‘и’ mean?

Response 2. Thank you for this remark. We agree with it. It is a typo.  On page 3 after formula 1 symbol ‘и’ is replaced with  ‘and’.

Reviewer 2 Report

Comments and Suggestions for Authors

In this article, the investigators propose criteria for distinguishing ideal from non-ideal behaviors of chemical systems based on the study of kinetics and the evaluation of time events, such as intersection points and CPE-extremum points. The conclusions are obtained by the careful study of simple kinetic models, one-step and two-step consecutive chemical reactions.

The results are well presented and documented and deserve publication. However, there are some technical questions the answer of which, in my opinion, will improve the article.

1. One such point is what we define as an ideal chemical model. The authors generally state, “Typically, the chemical ideality is verified assuming the weak interaction between molecules of the composition say under conditions the solution is diluted enough.”

Ideal models play a significant role in chemical thermodynamics and kinetics. The undergraduate literature describes gases and liquids in detail, as well as non-ideal models are introduced by invoking concepts such as fugacity and activity. It would be helpful and interesting enough if the authors gave more details on these concepts and their relation to the non-ideal generalized Marcelin-de Donde model (MDD).

2. On page 3 where the chemical reactions are described, the concentrations A(t) are … dimensionless, mol/s; What does it mean? dimensionless or mol/s? Explain.

3. The kinetics of ideal chemical systems are described by the Mass Action Law (MAL) but on page 3 the authors write “(ideal or non-ideal MAL)”. However, below it is written that “In the non-ideal kinetics of MDD”. Does it mean that MAL is valid for both ideal and non-ideal systems? This remark is related with the comment 1 and the definition of activities.

4. Page 3. “In the non-ideal system, all coefficients of non-ideality are equal 0, a_{jk} = 0, and g_j= 0.” . I should read “In an ideal system. …”

5. In the section of non-ideal two-step mechanism (pages 8-10) a lot of equations are written without any reference to their origin. For example, “(the cumbersome expression for D* is not presented).” Can one find it somewhere? Maybe, all these results are left as exercises to the reader! 

Some Spelling Corrections

1. Page 2, last sentence : … to analyze under …

2. Page 3, 6th Line and in other places : A Cyrillic symbol appears

3. Page 4, Line 1 : As is well known … Also, “As is indicated above, ...”

4. Page 4, Line 4 : “where А and B are reactants, initial reactant and product.”?

5. Page 4 : “the system is reversible or …”

Author Response

Response to Reviewer 2

In this article, the investigators propose criteria for distinguishing ideal from non-ideal behaviors of chemical systems based on the study of kinetics and the evaluation of time events, such as intersection points and CPE-extremum points. The conclusions are obtained by the careful study of simple kinetic models, one-step and two-step consecutive chemical reactions. The results are well presented and documented and deserve publication. However, there are some technical questions the answer of which, in my opinion, will improve the article.

Comment 1. “One such point is what we define as an ideal chemical model. The authors generally state, “Typically, the chemical ideality is verified assuming the weak interaction between molecules of the composition say under conditions the solution is diluted enough.”

Ideal models play a significant role in chemical thermodynamics and kinetics. The undergraduate literature describes gases and liquids in detail, as well as non-ideal models are introduced by invoking concepts such as fugacity and activity. It would be helpful and interesting enough if the authors gave more details on these concepts and their relation to the non-ideal generalized Marcelin-de Donde model (MDD). ”

Response 1. We decided to re-write some fragments of Section 2 “Kinetic Model” adding two sentences

Instead of

“In the ideal kinetics of MAL, the rate of a reversible isothermal reaction [1, 12, 17-20] is equal to ri = r+i - r-i, where r+i = k+i ПAj aij, r-i = k-i ПAjbij  - the rate of forward and reverse reactions, respectively, 1/s.

In the non-ideal kinetics of MDD which is a generalization of the ideal kinetics of MAL, the rate of any step of a reversible isothermal reaction [10-14, 18-20] is :

ri = ri0[exp(åj aijmj) - exp(åj bijmj)], i = 1, 2, …,                                             (4)

where ri0 > 0 - the rate, forward or reverse, under the equilibrium conditions, mol/(l s); µj - the pseudo-chemical potential (hereinafter referred to as the potential) of the reactant Aj, dimensionless:

mj = m0j + ln Aj  + gj, j = 1, 2, …,                                                         (5)

where m0j > 0 - constants; gj - correction terms for the non-ideality of reactants (gj = 0 for the ideal system); µj = µ*j/RT, µ*j - the chemical potential of the reactant Aj, J/mol; R is the universal gas constant, J/(mol oK); T - temperature, oK. “

the new text is written

“In our paper, the difference the ideal-system and non-ideal one is expressed in terms of chemical potentials.

In the ideal kinetics of MAL, the rate of a reversible isothermal reaction [1, 11, 17-20] is equal to ri = r+i - r-i, where r+i = k+i ПAj aij, r-i = k-i ПAjbij  - the rate of forward and reverse reactions, respectively, 1/s.

In the non-ideal kinetics of MDD which is a generalization of the ideal kinetics of MAL, the rate of any step of a reversible isothermal reaction [9-14, 18-20] is :

ri = ri0[exp(åj aijmj) - exp(åj bijmj)], i = 1, 2, …,                                             (4)

where ri0 > 0 - the rate, forward or reverse, under the equilibrium conditions, mol/(l s); µj - the pseudo-chemical potential (hereinafter referred to as the potential) of the reactant Aj, dimensionless:

mj = m0j + ln Aj  + gj, j = 1, 2, …,                                                         (5)

where m0j > 0 - constants; gj - correction terms for the non-ideality of reactants (gj = 0 for the ideal system); µj = µ*j/RT, µ*j - the chemical potential of the reactant Aj, J/mol; R is the universal gas constant, J/(mol oK); T - temperature, oK.

A goal of this paper is to find the difference between the ideal and non-ideal case based on kinetic behavior”

Added three links to original works by Marcelin and De-Donder:

  1. De Donder, Th.; Van Rysselberghe, P. Thermodynamic theory of affinity. London: Oxford Univ. Press, 1936.

136 p.

  1. Van Rysselberghe, P., Reaction rates and affinities. J. Chem. Phys., 29, 640-642 (1958)
  2. Feinberg, M. On chemical kinetics of a certain class. Arch. Rat. Mechanics and Analysis. 1972, 46, 1-41.

Comments 2. On page 3 where the chemical reactions are described, the concentrations A(t) are … dimensionless, mol/s; What does it mean? dimensionless or mol/s? Explain.

Response 2. After formula (2) the expression “where Aj = Aj(t) - reactant concentrations, dimensionless, mol/l;” заменено на “where Aj = Aj(t) - reactant concentrations, mol/l;”

Comments 3. The kinetics of ideal chemical systems are described by the Mass Action Law (MAL) but on page 3 the authors write “(ideal or non-ideal MAL)”. However, below it is written that “In the non-ideal kinetics of MDD”. Does it mean that MAL is valid for both ideal and non-ideal systems? This remark is related with the comment 1 and the definition of activities.

Response 3. After formula (2), the expression “ri - the rate of any step (ideal or non-ideal MAL) that satisfies the natural requirement (ri = 0 at Aj = 0), mol/(l s);” is replaced by

ri - the rate of any step reaction (ideal or non-ideal), that satisfies the natural requirement (ri = 0 at Aj = 0), mol/(l s);”. The kinetics of MAL is valid only for ideal systems, as stated above in Response 1.

Comments 4. Page 3. “In the non-ideal system, all coefficients of non-ideality are equal 0, a_{jk} = 0, and g_j= 0.” . I should read “In an ideal system. …”

Response 4. Yes, that's right, thank you for your attention. The phrase "In the non-ideal system" has been replaced by " In an ideal system.”

Comments 5. In the section of non-ideal two-step mechanism (pages 8-10) a lot of equations are written without any reference to their origin. For example, “(the cumbersome expression for D* is not presented).” Can one find it somewhere? Maybe, all these results are left as exercises to the reader! 

Response 5. The equations on pages (8)-(10) are necessary for describing the non-ideal kinetic model (Subsection 4.2.). The section “Kinetic Model” provides the definition of the linear relaxation time t = 1/minïRe ljï, [23-26], where lj - eigenvalues. In the non-ideal system (19), the eigenvalues l*1,2 = (s* ± D*)/2, where s* = trace(J), D* = det(J) – are the characteristics of the Jacobian, trace and determinant respectively and D* = (s*2 – 4D*)1/2. The expression for D* is not presented, since it is too big for this paper. However, readers can easily derive this equation using the computer algebra methods from the MATLAB.

Comments 6. Some Spelling Corrections

  1. Page 2, last sentence : … to analyze under …
  2. Page 3, 6th Line and in other places : A Cyrillic symbol appears
  3. Page 4, Line 1 : As is well known … Also, “As is indicated above, ...”
  4. Page 4, Line 4 : “where А and B are reactants, initial reactant and product.”?
  5. Page 4 : “the system is reversible or …”

Response 6. Items 1, 2, 3, 5 - corrected in accordance with the reviewer's recommendations.

Item 4 – left unchanged (Section 3), since in our opinion it does not require adjustment.